# Gradient Estimation with Discrete Stein Operators

**Jiaxin Shi**[†]
Stanford University
jiaxins@stanford.edu

**Yuhao Zhou**
Tsinghua University
yuhaoz.cs@gmail.com

**Jessica Hwang**
Stanford University
jjhwang@stanford.edu

**Michalis K. Titsias**
DeepMind
mtitsias@google.com

**Lester Mackey**
Microsoft Research New England
lmackey@microsoft.com

## Abstract

Gradient estimation—approximating the gradient of an expectation with respect to the parameters of a distribution—is central to the solution of many machine learning problems. However, when the distribution is discrete, most common gradient estimators suffer from excessive variance. To improve the quality of gradient estimation, we introduce a variance reduction technique based on Stein operators for discrete distributions. We then use this technique to build flexible control variates for the REINFORCE leave-one-out estimator. Our control variates can be adapted online to minimize variance and do not require extra evaluations of the target function. In benchmark generative modeling tasks such as training binary variational autoencoders, our gradient estimator achieves substantially lower variance than state-of-the-art estimators with the same number of function evaluations.

## 1 Introduction

Modern machine learning relies heavily on gradient methods to optimize the parameters of a learning system. However, exact gradient computation is often difficult. For example, in variational inference for training latent variable models [29, 43], policy gradient algorithms in reinforcement learning [65], and combinatorial optimization [40], the exact gradient features an intractable sum or integral introduced by an expectation under an evolving probability distribution. To make progress, one resorts to estimating the gradient by drawing samples from that distribution [39, 50].

The two main classes of gradient estimators used in machine learning are the pathwise or reparameterization gradient estimators [29, 47, 58] and the REINFORCE or score function estimators [18, 65]. The pathwise estimators have shown great success in training variational autoencoders [29] but are only applicable to continuous probability distributions. The REINFORCE estimators are more general-purpose and easily accommodate discrete distributions but often suffer from excessively high variance.

To improve the quality of REINFORCE estimators, we develop a new variance reduction technique for discrete expectations. Our method is based upon Stein operators [see, e.g., 1, 55], computable functionals that generate mean-zero functions under a target distribution and provide a natural way of designing control variates (CVs) for stochastic estimation.

We first provide a general recipe for constructing practical Stein operators for discrete distributions (Table 1), generalizing the prior literature [6, 8, 9, 16, 25, 46, 67]. We then develop a gradient estimation framework—RODEO—that augments REINFORCE estimators with mean-zero CVs generated from Stein operators. Finally, inspired by Double CV [60], we extend our method to develop CVs for REINFORCE leave-one-out estimators [30, 49] to further reduce the variance.

---

[†]Work done while at Microsoft Research New England.

36th Conference on Neural Information Processing Systems (NeurIPS 2022).

Table 1: Discrete Stein operators that generate mean-zero functions under $q$ (i.e., $\mathbb{E}_q[(Ah)(x)] = 0$).

| Stein Operator | $(Ah)(x)$ |
|---|---|
| Gibbs (4) | $\frac{1}{d} \sum_{i=1}^d \sum_{y_{-i}=x_{-i}} q(y_i\lvert x_{-i}) h(y) - h(x)$ |
| MPF (6) | $\sum_{y\in\mathcal{N}_x, y\neq x} \sqrt{q(y)/q(x)}(h(y) - h(x))$ |
| Barker (6) | $\sum_{y\in\mathcal{N}_x, y\neq x} \frac{q(y)}{q(x)+q(y)}(h(y) - h(x))$ |
| Difference (8) | $\frac{1}{d} \sum_{i=1}^d h(\mathtt{dec}_i(x)) - \frac{q(\mathtt{inc}_i(x))}{q(x)} h(x)$ |

The benefits of using Stein operators to construct discrete CVs are twofold. First, the operator structure permits us to learn CVs with a flexible functional form such as those parameterized by neural networks. Second, since our operators are derived from Markov chains on the discrete support, they naturally incorporate information from neighboring states of the process for variance reduction.

We evaluate RODEO on 15 benchmark tasks, including training binary variational autoencoders (VAEs) with one or more stochastic layers. In most cases and with the same number of function evaluations, RODEO delivers lower variance and better training objectives than the state-of-the-art gradient estimators DisARM [14, 69], ARMS [13], Double CV [60], and RELAX [20].

## 2 Background

We consider the problem of maximizing the objective function $\mathbb{E}_{q_\eta}[f(x)]$ with respect to the parameters $\eta$ of a discrete distribution $q_\eta(x)$. Throughout the paper we assume $f(x)$ is a differentiable function of real-valued inputs $x \in \mathbb{R}^d$ but is only evaluated at a discrete subset $\mathcal{X}^d$ due to the discreteness of $q_\eta$.[2] Exact computation of the expectation is typically intractable due to the complex nature of $f(x)$ and $q_\eta(x)$. The standard workaround is to rewrite the gradient as $\nabla_\eta \mathbb{E}_{q_\eta}[f(x)] = \mathbb{E}_{q_\eta}[f(x)\nabla_\eta \log q_\eta(x)]$ and employ the Monte Carlo gradient estimator known as the score function or REINFORCE estimator [18, 65]:

$$\frac{1}{K}\sum_{k=1}^K (f(x^{(k)}) - b)\nabla_\eta \log q_\eta(x^{(k)}) \quad \text{for} \quad x^{(1)},\dots,x^{(K)} \overset{i.i.d.}{\sim} q_\eta. \quad \text{(REINFORCE)}$$

Here, $b$ is a constant called the "baseline" introduced to reduce the variance of the estimator by reducing the scaling effect of $f(x^{(k)})$. Since $\mathbb{E}_{q_\eta}[\nabla_\eta \log q_\eta(x)] = 0$, the REINFORCE estimator is unbiased for any choice of $b$, and the term $b\nabla_\eta \log q_\eta(x)$ is known as a *control variate* (CV) [42, Ch. 8]. The optimal baseline can be estimated using additional function evaluations [7, 45, 64]. A simpler approach is to use moving averages of $f$ from historical evaluations or to train function approximators to mimic those values [37]. When $K \geq 2$, a powerful variant of REINFORCE is obtained by replacing $b$ with the leave-one-out average of function values:

$$\frac{1}{K}\sum_{k=1}^K \left(f(x^{(k)}) - \frac{1}{K-1}\sum_{j\neq k} f(x^{(j)})\right)\nabla_\eta \log q_\eta(x^{(k)}). \quad \text{(RLOO)}$$

This approach is often called the REINFORCE leave-one-out (RLOO) estimator [30, 48, 49] and was recently observed to have very strong performance in training discrete latent variable models [14, 48].

All the above methods construct a baseline that is independent of the point $x^{(k)}$ under consideration, but there are other ways to preserve the unbiasedness of the estimator. We are free to use a sample-dependent baseline $b(x^{(k)})$ as long as the expectation $c \triangleq \mathbb{E}_{q_\eta}[b(x^{(k)})\nabla_\eta \log q_\eta(x^{(k)})]$ is easily computed or has a low-variance unbiased estimate. In this case, we can correct for any bias introduced by $b(x^{(k)})$ by adding in this expectation: $\frac{1}{K}\sum_{k=1}^K (f(x^{(k)}) - b(x^{(k)}))\nabla_\eta \log q_\eta(x^{(k)}) + c$. For example, $b(x^{(k)})$ can be a lower bound or a Taylor expansion of $f(x^{(k)})$ [22, 43]. Taking this a step further, the Double CV estimator [60] proposed to treat $f(x^{(k)}) - b(x^{(k)})$ as the effective objective function and apply the leave-one-out idea:

$$\frac{1}{K}\sum_{k=1}^K \left[\left(f(x^{(k)}) - b_k(x^{(k)})\right) - \frac{1}{K-1}\sum_{j\neq k}\left(f(x^{(j)}) - b_j(x^{(j)})\right)\right]\nabla_\eta \log q_\eta(x^{(k)}) + c.$$

The resulting estimator adds two CVs to RLOO: the *global* CV $b_k(x^{(k)})\nabla_\eta \log q_\eta(x^{(k)})$ and the *local* CV $b_k(x^{(j)})$. Intuitively, $b_k(x^{(j)})$ is aimed at reducing the variance of the LOO average. This is

---

[2]This assumption holds for many discrete probabilistic models [21] including the binary VAEs of Section 6.

motivated by the fact that replacing $\frac{1}{K-1}\sum_{j\neq k}f(x^{(j)})$ with $\mathbb{E}_{q_\eta}[f]$ in RLOO leads to lower variance [60, Prop 1]. To obtain a tractable correction term $c$, Titsias and Shi [60] adopt a linear design of $b_k$: $b_k(x) = \alpha \cdot \frac{1}{K-1}\sum_{j\neq k}f(x_j)(x-\mu)$ for $\mu = \mathbb{E}_{q_\eta}[x]$ and a coefficient $\alpha$.

Although the Double CV framework points to a promising new direction for developing better REINFORCE estimators, one only obtains significant reduction in variance when $b_k$ is strongly correlated with $f$. This may fail to hold for the above linear $b_k$ and a highly nonlinear $f$, especially in applications like training deep generative models.

In the following two sections, we will introduce a method that allows us to use very flexible CVs while still maintaining a tractable correction term. Our method enables online adaptation of CVs to minimize gradient variance (similar to RELAX [20]) but does not assume $q_\eta$ has a continuous reparameterization. We then apply it to generalize the linear CVs in Double CV to very flexible ones such as neural networks. Moreover, we provide an effective CV design based on surrogate functions that requires no additional evaluation of $f$ compared to RLOO.

## 3 Control Variates from Discrete Stein Operators

At the heart of our new estimator is a technique for generating flexible discrete CVs, that is for generating a rich class of functions that have known expectations under a given discrete distribution $q$. One way to achieve this is to identify any discrete-time Markov chain $(X^{(t)})_{t=0}^\infty$ with stationary distribution $q$. Then, the transition matrix $P$, defined via $P_{xy} = P(X^{(t+1)} = y | X^{(t)} = x)$, satisfies $P^\top q = q$ and hence

$$\mathbb{E}_q[(P-I)h] = 0, \tag{1}$$

for any integrable function $h$. We overload our notation so that, for any suitable matrix $A$, $(Ah)(x) \triangleq \sum_y A_{xy}h(y)$. In other words, for any integrable $h$, the function $(P-I)h$ is a valid CV as it has known mean under $q$. Moreover, the linear operator $P - I$ is an example of a *Stein operator* [1, 55] in the sense that it generates mean-zero functions under $q$. In fact, both Stein et al. [56] and Dellaportas and Kontoyiannis [12] developed CVs of the form $(P-I)h$ based on reversible discrete-time Markov chains and linear input functions $h(x) = x_i$.

More generally, Barbour [3] and Henderson [23] observed that if we identify a continuous-time Markov chain $(X^{(t)})_{t\geq 0}$ with $q$ as its stationary distribution, then the *generator* $A$, defined via

$$(Ah)(x) = \lim_{t\to 0}\frac{\mathbb{E}[h(X^{(t)})|X^{(0)}=x]-h(x)}{t}, \tag{2}$$

satisfies $A^\top q = 0$ and hence $\mathbb{E}_q[Ah] = 0$ for all integrable $h$. Therefore, $A$ is also a Stein operator suitable for generating CVs. Moreover, since any discrete-time chain with transition matrix $P$ can be embedded into a continuous-time chain with transition rate matrix $A = P - I$, this continuous-time construction is strictly more general [57, Ch. 4].

### 3.1 Discrete Stein operators

We next present several examples of broadly applicable discrete Stein operators (summarized in Table 1) that can serve as practical defaults.

**Gibbs Stein operator**   The transition kernel of the random-scan Gibbs sampler with stationary distribution $q$ [see, e.g., 17] is

$$P_{xy} = \frac{1}{d}\sum_{i=1}^d q(y_i|x_{-i})\mathbf{1}(y_{-i} = x_{-i}), \tag{3}$$

where $\mathbf{1}(\cdot)$ is the indicator function. The associated Stein operator is $A = P - I$ with

$$(Ah)(x) = \frac{1}{d}\sum_{i=1}^d \sum_{y_{-i}=x_{-i}} q(y_i|x_{-i})h(y) - h(x). \tag{4}$$

In the binary variable setting, (4) recovers the operator Bresler and Nagaraj [8], Reinert and Ross [46] used to bound distances between the stationary distributions of Glauber dynamics Markov chains.

**Zanella Stein operator**   Zanella [70] studied continuous-time Markov chains with generator

$$A_{xy} = \kappa\left(q(y)/q(x)\right)\mathbf{1}(y \in \mathcal{N}_x, y \neq x) - \sum_{z\neq x}A_{xz}\mathbf{1}(y = x), \tag{5}$$

where $\mathcal{N}_x$ denotes the transition neighborhood of $x$ and $\kappa$ is a continuous positive function satisfying $\kappa(t) = t\kappa(1/t)$. Conveniently, the neighborhood structure can be arbitrarily sparse. Moreover, the $\kappa$ constraint ensures that the chain satisfies detailed balance (i.e., $q(x)A_{xy} = q(y)A_{yx}$) and hence that $A^\top q = 0$. Hodgkinson et al. [24] call the associated operator the *Zanella Stein operator*:

$$(Ah)(x) = \sum_{y \in \mathcal{N}_x, y \neq x} \kappa\left(\tfrac{q(y)}{q(x)}\right)(h(y) - h(x)). \tag{6}$$

Important special cases include the *minimum probability flow [53] (MPF) Stein operator* ($\kappa(t) = \sqrt{t}$) discussed in Barp et al. [6] and the *Barker [4] Stein operator* ($\kappa(t) = \frac{t}{t+1}$).

**Birth-death Stein operator**  Let $e_1, \ldots, e_d$ represent the standard basis vectors on $\mathbb{R}^d$. For any finite-cardinality space, we may index the elements of $\mathcal{X} = \{s_0, \ldots, s_{m-1}\}$, let $\mathtt{idx}(x_i)$ return the index of the element that $x_i$ represents, and define the increment and decrement operators

$$\mathtt{inc}_i(x) = x + e_i(s_{(\mathtt{idx}(x_i)+1) \bmod m} - x_i) \quad \text{and} \quad \mathtt{dec}_i(x) = x + e_i(s_{(\mathtt{idx}(x_i)-1) \bmod m} - x_i).$$

Then the product *birth-death process* [28] on $\mathcal{X}^d$ with birth rates $b_{i,x} = \frac{q(\mathtt{inc}_i(x))}{q(x)}$, death rates $d_{i,x} = 1$, and generator

$$A_{xy} = \tfrac{1}{d} \sum_{i=1}^d b_{i,x} \mathbf{1}(y = \mathtt{inc}_i(x)) + d_{i,x} \mathbf{1}(y = \mathtt{dec}_i(x)) - (b_{i,x} + d_{i,x})\mathbf{1}(y = x),$$

has $q$ as a stationary distribution, as $A^\top q = 0$. This construction yields the birth-death Stein operator

$$(Ag)(x) = \tfrac{1}{d} \sum_{i=1}^d b_{i,x}(g(\mathtt{inc}_i(x)) - g(x)) - d_{i,x}(g(x) - g(\mathtt{dec}_i(x))). \tag{7}$$

An analogous operator is available for countably infinite $\mathcal{X}$ [24], and Brown and Xia [9], Eichelsbacher and Reinert [16], Holmes [25] used related operators to characterize convergence to discrete target distributions. Moreover, by substituting $h(x) = g(x) - g(\mathtt{inc}(x))$ in (7), we recover the *difference Stein operator* proposed by Yang et al. [67] without reference to birth-death processes:

$$(Ah)(x) = \tfrac{1}{d} \sum_{i=1}^d h(\mathtt{dec}_i(x)) - b_{i,x}h(x). \tag{8}$$

**Choosing a Stein operator**  Despite being better-known, the difference and MPF operators often suffer from large variance and numerical instability due to their use of unbounded probability ratios $q(y)/q(x)$. As a result, we recommend the numerically stable Gibbs operator when each component of $x$ is binary or takes on a small number of values. When $x_i$ takes on a large number of values ($m$), the Gibbs operator suffers from linear scaling with $m$. In this case, we recommend the Barker operator where a sparse neighborhood structure can be specified, such as $\mathcal{N}_x = \{\mathtt{inc}_i(x), \mathtt{dec}_i(x) \text{ for } i \in [d]\}$. The Barker operator is numerically stable as its $\kappa(\frac{q(y)}{q(x)}) = \frac{q(y)}{q(x)+q(y)}$.

## 4 Gradient Estimation with Discrete Stein Operators

Recall that REINFORCE estimates the gradient $\mathbb{E}_{q_\eta}[f(x)\nabla_\eta \log q_\eta(x)]$. Due to the existence of $\nabla_\eta \log q_\eta(x)$ as a weighting function, we apply a discrete Stein operator to a vector-valued function $\tilde{h} : \mathcal{X} \to \mathbb{R}^d$ per dimension to construct the following estimator with a mean-zero CV:

$$\mathbb{E}_{q_\eta}[f(x)\nabla_{\eta_i} \log q_\eta(x) + (A\tilde{h}_i)(x)]. \tag{9}$$

Ideally, we want to choose the $\tilde{h}$ such that $A\tilde{h}_i$ will be strongly correlated with $f(x)\nabla_{\eta_i} \log q_\eta(x)$ to reduce its variance. The optimal $\tilde{h}_i$ is given by the solution of Poisson equation

$$\mathbb{E}_{q_\eta}[f\nabla_{\eta_i} \log q_\eta] - f\nabla_{\eta_i} \log q_\eta = A\tilde{h}_i. \tag{10}$$

We could learn a separate $\tilde{h}_i$ per dimension to approximate the solution of (10). However, this poses a difficult optimization problem that requires simultaneously solving $d$ Poisson equations. Instead, we will incorporate knowledge about the structure of the solution to simplify the optimization.

To determine a candidate functional form for $\tilde{h}$, we draw inspiration from an "optimal" Markov chain in which the current state is ignored entirely and the new state is generated independently from $q_\eta$. In

---

**Algorithm 1** Optimizing $\mathbb{E}_{q_\eta}[f_\theta(x)]$ with RODEO gradients

---

    **input:** Objective $f_\theta$, sample points $x^{(1:K)} \stackrel{i.i.d.}{\sim} q_\eta$, Stein operator $A$, step sizes $\alpha_t, \beta_t$
    **for** $t = 1 : T$ **do**
1:  $\{f_\theta(x^{(k)}), \nabla_\theta f_\theta(x^{(k)}), \nabla f_\theta(x^{(k)})\}_{k=1}^K \leftarrow \text{autodiff}(f_\theta, x^{(1:K)})$.
2:  Compute the surrogates $h_k(x^{(j)}), h_k^\star(x^{(j)})$ of (13), (14) **for** $j \neq k$ and $j, k = 1, \cdots, K$.
3:  Compute the RODEO gradient estimator $g_\gamma(x^{(1:K)})$.
4:  $\theta \leftarrow \theta + \alpha_t \frac{1}{K} \sum_{k=1}^K \nabla_\theta f(x^{(k)})$.
5:  $\eta \leftarrow \eta + \alpha_t g_\gamma(x^{(1:K)})$.
6:  Update hyperparameters: $\gamma \leftarrow \gamma - \beta_t \nabla_\gamma \|g_\gamma(x^{(1:K)})\|_2^2$.

---

this case, $(P - I)\tilde{h}_i$ becomes $\mathbb{E}_{q_\eta}[\tilde{h}_i] - \tilde{h}_i$, and the optimal solution is $\tilde{h}_i = f\nabla_{\eta_i} \log q_\eta$. Inspired by this, we consider $\tilde{h}$ of the form

$$\tilde{h}(x) = h(x)\nabla_\eta \log q_\eta(x), \tag{11}$$

where we now only need to learn a scalar-valued function $h$. Notably, when $h$ exactly equals $f$ and $A = P - I$ for any discrete time Markov chain kernel $P$, our CV adjustment amounts to Rao-Blackwellization [42, Sec. 8.7], as we end up replacing $f(x)\nabla_{\eta_i} \log q_\eta(x)$ with its conditional expectation $P(f\nabla_{\eta_i} \log q_\eta)(x) = \mathbb{E}_{X_{t+1}|X_t=x}[f(X_{t+1})\nabla_{\eta_i} \log q_\eta(X_{t+1})]$. This yields a guaranteed variance reduction.

**Surrogate function design**   Based on the above reasoning, we can view $h$ as a surrogate for $f$. We avoid directly setting $h = f$ because our Stein operators evaluate $h$ at all neighbors of the sample points, and $f$ can be very expensive to evaluate [see, e.g., 51]. To avoid this problem, we first observe that $\tilde{h}$ (11) can be made sample-specific, i.e., we can use a different $\tilde{h}_k$ for each sample point $x^{(k)}$:

$$\tfrac{1}{K} \sum_{k=1}^K [f(x^{(k)})\nabla_\eta \log q_\eta(x^{(k)}) + (A\tilde{h}_k)(x^{(k)})]. \tag{12}$$

We then consider the following choices of $h_k$ that are informed about $f$ while being cheap to evaluate:

$$h_k(y) = \tfrac{1}{K-1} \sum_{j \neq k} H(f(x^{(j)}), \nabla f(x^{(j)})^\top(y - x^{(j)})). \tag{13}$$

Here $H$ belongs to a flexible parametric family of functions such as neural networks and is chosen to have significantly lower cost than $f$. $y$ will take on the values of $x^{(k)}$ and its neighbors in the Markov chain. We omit $x^{(k)}$ in the sum (13) to ensure that the function $h_k$ is independent of $x^{(k)}$ and hence that $\mathbb{E}_{q_\eta}[\frac{1}{K} \sum_{k=1}^K (A\tilde{h}_k)(x^{(k)})] = 0$. Moreover, this surrogate function design introduces no additional evaluations of $f$ beyond those required for the usual RLOO estimator. Also, as observed by Titsias and Shi [60], for many applications, including VAE training, where $f$ has parameters $\theta$ learned through gradient-based optimization, $\{\nabla f(x^{(k)})\}_{k=1}^K$ can be obtained "for free" from the same backpropagation used to compute $\nabla_\theta f_\theta(x^{(k)})$ (see Algorithm 1).

**RODEO**   We can further improve the estimator by leveraging discrete Stein operators and the above surrogate function design to construct both the global and local CVs in the Double CV framework [60] (see Section 2). We call our final estimator *RODEO* for *R*L*O*O with *D*iscrete St*E*in *O*perators:

$$\tfrac{1}{K} \sum_{k=1}^K [(f(x^{(k)}) - \tfrac{1}{K-1} \sum_{j \neq k} (f(x^{(j)}) + (\boldsymbol{Ah_j})(\boldsymbol{x^{(j)}})))\nabla_\eta \log q_\eta(x^{(k)}) + (\boldsymbol{A\tilde{h}_k^\star})(\boldsymbol{x^{(k)}})], \quad \text{(RODEO)}$$

where $\tilde{h}_k^\star(y) = h_k^\star(y)\nabla_\eta \log q_\eta(y)$ and $\{h_k, h_k^\star\}_{k=1}^K$ are scalar-valued functions. Here, $(Ah_j)(x^{(j)})$ is a scalar-valued CV introduced to reduce the variance of the leave-one-out baseline $\frac{1}{K-1} \sum_{j \neq k} f(x^{(j)})$, while $(A\tilde{h}_k^\star)(x^{(k)})$ acts as a global CV to further reduce the variance of the gradient estimate. We adopt the aforementioned design of $h$ (13) and $\tilde{h}$ (11) for the two CVs. In Appendix B, we show that the RODEO estimator is unbiased for $\nabla_\eta \mathbb{E}_{q_\eta}[f(x)]$ when each $h_k$ is defined as in (13) and

$$h_k^\star(y) = \tfrac{1}{K-1} \sum_{j \neq k} H^\star(f(x^{(j)}), \nabla f(x^{(j)})^\top(y - x^{(j)})). \tag{14}$$

**Optimization with RODEO**   In practice, we let the functions $H$ and $H^\star$ share a neural network architecture with two output units. Since the estimator is unbiased, we can optimize the

Table 2: Training binary latent VAEs with $K = 2, 3$ (except for RELAX which uses 3 evaluations) on MNIST, Fashion-MNIST, and Omniglot. We report the average ELBO ($\pm 1$ standard error) on the training set after 1M steps over 5 independent runs. Test data bounds are reported in Table 4.

| | | Bernoulli Likelihoods | | | Gaussian Likelihoods | | |
|---|---|---|---|---|---|---|---|
| | | MNIST | Fashion-MNIST | Omniglot | MNIST | Fashion-MNIST | Omniglot |
| $K = 2$ | DisARM | $-102.75 \pm 0.08$ | $-237.68 \pm 0.13$ | $-116.50 \pm 0.04$ | $668.03 \pm 0.61$ | $182.65 \pm 0.47$ | $446.61 \pm 1.16$ |
| | Double CV | $-102.14 \pm 0.06$ | $-237.55 \pm 0.16$ | $-116.39 \pm 0.10$ | $676.87 \pm 1.18$ | $186.35 \pm 0.64$ | $447.65 \pm 0.87$ |
| | RODEO (Ours) | $\mathbf{-101.89 \pm 0.17}$ | $\mathbf{-237.44 \pm 0.09}$ | $\mathbf{-115.93 \pm 0.06}$ | $\mathbf{681.95 \pm 0.37}$ | $\mathbf{191.81 \pm 0.67}$ | $\mathbf{454.74 \pm 1.11}$ |
| $K = 3$ | ARMS | $-100.84 \pm 0.14$ | $-237.05 \pm 0.12$ | $-115.21 \pm 0.07$ | $683.55 \pm 1.01$ | $193.07 \pm 0.34$ | $457.98 \pm 1.03$ |
| | Double CV | $-100.94 \pm 0.09$ | $-237.40 \pm 0.11$ | $-115.06 \pm 0.12$ | $686.48 \pm 0.68$ | $193.93 \pm 0.20$ | $457.44 \pm 0.79$ |
| | RODEO (Ours) | $\mathbf{-100.46 \pm 0.13}$ | $\mathbf{-236.88 \pm 0.12}$ | $\mathbf{-115.01 \pm 0.05}$ | $\mathbf{692.37 \pm 0.39}$ | $\mathbf{196.56 \pm 0.42}$ | $461.87 \pm 0.90$ |
| | RELAX (3 evals) | $-101.99 \pm 0.04$ | $-237.74 \pm 0.12$ | $-115.70 \pm 0.08$ | $688.58 \pm 0.52$ | $196.38 \pm 0.66$ | $\mathbf{462.23 \pm 0.63}$ |

parameters $\gamma$ of the network in an online fashion to minimize the variance of the estimator (similar to [20]). Specifically, if we denote the RODEO gradient estimate by $g_\gamma(x^{(1:K)})$, then $\nabla_\gamma \text{Tr}(\text{Var}(g_\gamma(x^{(1:K)}))) = \mathbb{E}[\nabla_\gamma \|g_\gamma(x^{(1:K)})\|_2^2]$. In Algorithm 1, we use an unbiased estimate of this hyperparameter gradient, $\nabla_\gamma \|g_\gamma(x^{(1:K)})\|_2^2$, to update $\gamma$ after each optimization step of $\eta$.

## 5 Related Work

As we have seen in Section 2, there is a long history of designing CVs for REINFORCE estimators using "baselines" [7, 37, 45, 64]. Recent progress is mostly driven by leave-one-out [30, 38, 48, 49] and sample-dependent baselines [20, 22, 43, 60, 62]. REBAR [62] constructs the baseline through continuous relaxation of the discrete distribution [26, 35] and applies reparameterization gradients to the correction term. As a result REBAR uses three evaluations of $f$ for each $x^{(k)}$ instead of usual single evaluation (see Appendix A for a detailed explanation). The RELAX [20] estimator generalizes REBAR by noticing that their continuous relaxation can be replaced with a free-form CV. However, in order to get strong performance, RELAX still includes the continuous relaxation in their CV and only adds a small deviation to it. Therefore, RELAX also uses three evaluations of $f$ for each $x^{(k)}$ and is usually considered more expensive than other estimators.

An attractive property of the RODEO estimator is that it incorporates information from neighboring states thanks to the Stein operator while avoiding additional costly $f$ evaluations thanks to the learned surrogate functions $h_k$. Estimators based on analytic local expectation [59, 61] and GO gradients [11] also use neighborhood information but only at the cost of many additional target function evaluations. In fact, the local expectation gradient [59] can be viewed as a Stein CV adjustment RODEO with a Gibbs Stein operator and the target function $f$ used directly instead of the surrogate $h$. The downside of these approaches is that $f$ must be evaluated $Kd$ times per training step instead of $K$ times as in RODEO, a prohibitive cost when $f$ is expensive and $d \geq 200$ as in Section 6.

Prior work has also studied variance reduction methods based on sampling without replacement [31] and antithetic sampling [13–15, 68] for gradient estimation. DisARM [14, 69] was shown to outperform RLOO estimators when $K = 2$ and ARMS [13] generalizes DisARM to $K > 2$.

Stein operators for continuous distributions have also been leveraged for effective variance reduction in a variety of learning tasks [2, 5, 36, 41, 52, 54] including gradient estimation [27, 34]. In particular, the gradient estimator of Liu et al. [34] is based on the Langevin Stein operator [19] for continuous distributions and coincides with the continuous counterpart of RELAX [20]. In contrast, our approach considers discrete Stein operators for Monte Carlo estimation in discrete distributions with exponentially large state spaces. Recently, Parmas and Sugiyama [44, App. E.4] used a probability flow perspective to characterize *all* unbiased gradient estimators satisfying a mild technical condition; our estimators fall into this broad class but were not specifically investigated.

## 6 Experiments

Python code replicating all experiments can be found at https://github.com/thjashin/rodeo.

## 6.1 Training Bernoulli VAEs

Following Dong et al. [14] and Titsias and Shi [60], we conduct experiments on training variational auto-encoders [29, 47] (VAEs) with Bernoulli latent variables. VAEs are models with a joint density $p_\theta(y, x)$, where $x$ is the latent variable and $\theta$ denotes model parameters. They are typically learned through maximizing the *evidence lower bound* (ELBO) $\mathbb{E}_{q_\eta(x|y)}[f(x)]$ for an auxiliary inference network $q_\eta(x|y)$ and $f(x) \triangleq \log p_\theta(y, x) - \log q_\eta(x|y)$. In our experiments, $p(x)$ and $q_\eta(x|y)$ are high-dimensional distributions where each dimension of the random variable is an independent Bernoulli. Since exact gradient computations are intractable, we will use gradient estimators to learn the parameters $\eta$ of the inference network. The VAE architecture and training experimental setup follows Titsias and Shi [60], and details are given in Appendix D. The dimensionality of the latent variable $x$ is $d = 200$. The functions $H$ (13) and $H^*$ (14) share a neural network architecture with two output units and a single hidden layer with 100 units. For the numerical stability and variance reasons discussed in Section 3, we use the Gibbs Stein operator (4) as a default choice in our experiments, but we revisit this choice in Section 6.3.

We consider the MNIST [33], Fashion-MNIST [66] and Omniglot [32] datasets using their standard train, validation, and test splits. We use both binary and continuous VAE outputs ($y$) as in Titsias and Shi [60]. In the binary output setting, data are dynamically binarized, and the Bernoulli likelihood is used; in the continuous output setting, data are centered between $[-1, 1]$, and the Gaussian likelihood with learnable diagonal covariance parameters is used.

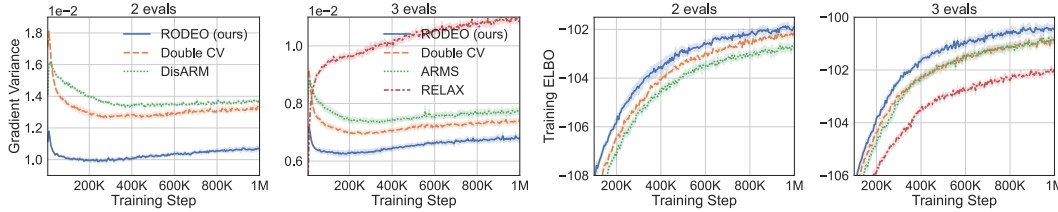

Figure 1: Training binary latent VAEs with 2 or 3 $f$ evaluations per step on binarized MNIST.

**$K = 2$**  In the first set of experiments we focus on the most common setting of $K = 2$ sample points and compare our variance reduction method to its counterparts including DisARM [14] and Double CV [60]. Results for RLOO are omitted since it is consistently outperformed by Double CV [see 60]. Table 2 shows, on all three datasets, RODEO achieves the best training ELBOs. In Figure 1 (left), we plot the gradient variance and average training ELBOs against training steps for all estimators on dynamically binarized MNIST. RODEO outperforms DisARM and Double CV by a large margin in gradient variance.

Next, we consider VAEs with Gaussian likelihoods trained on non-binarized datasets. In this case the gradient estimates suffer from even higher variance due to the large range of values $f(x)$ can take. The results are plotted in Figure 2. We see that RODEO has substantially lower variance than DisARM and Double CV, leading to significant improvements in training ELBOs. In Appendix Figure 7, we find that RODEO also consistently yields the best test set performance in all six settings.

**$K = 3$**  In the second set of experiments we compare RELAX [20], which uses three evaluations of $f$ per training step, with RODEO, Double CV, and ARMS [13] for $K = 3$. Figure 1 (right) and Table 2 demonstrate that RODEO outperforms the three previous methods and generally leads to the lowest variance. Although RELAX was often observed to have very strong performance in prior work [14, 60], our results in Figure 1 suggest that, for dynamically binarized datasets, much larger gains can be achieved by using the same number of function evaluations in other estimators.

For the experiments mentioned above, we report final training ELBOs in Table 2, test log-likelihood bounds in Appendix Table 4, and binarized MNIST average running time in Appendix Table 5. For $K = 3$, RODEO has the best final performance in 5 out of 6 tasks and runtime nearly identical to RELAX. For $K = 2$, RODEO has the best final performance for all tasks and is 1.5 times slower than Double CV and DisARM. We attribute the runtime gap to the Gibbs operator (4) which performs $2d$ evaluations of the auxiliary functions $H$ and $H^*$ in (13) and (14). While this results in some extra cost, the network parameterizing $H$ and $H^*$ takes only 2-dimensional inputs, produces scalar outputs, and is typically small relative to the VAE model. As a result, the evaluation of $H$ and $H^*$ is

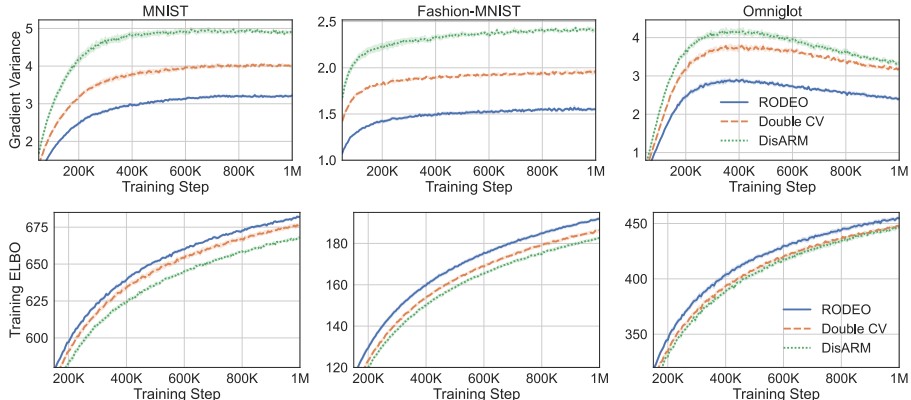

Figure 2: Training binary latent VAEs with Gaussian likelihoods, $K = 2$, and non-binarized datasets.

significantly cheaper than that of $f$, and its relative contribution to runtime shrinks as the cost of $f$ grows. To demonstrate this, we include a wall clock time comparison of our method with RLOO in Appendix C.1. In this experiment we replace the two-layer MLP-based VAE with a ResNet VAE, where the cost of $f$ is significantly higher than the single-layer MLP of $H, H^*$. In this case, RODEO and RLOO have very close per-iteration time (0.025s vs. 0.023s). And RODEO achieves better training ELBOs than RLOO for the same amount of time.

## 6.2 Training hierarchical Bernoulli VAEs

To investigate the performance of RODEO when scaling to hierarchical discrete latent variable models, we follow DisARM [14, 68] to train VAEs with 2/3/4 stochastic layers, each of which consists of 200 Bernoulli variables. We set $K = 2$ and compare our estimator with DisARM and Double CV on dynamically binarized MNIST, Fashion-MNIST, and Omniglot. For each stochastic layer, we use a different CV network which has the same architecture as those in our VAE experiments from the previous section. More details are presented in Appendix D.

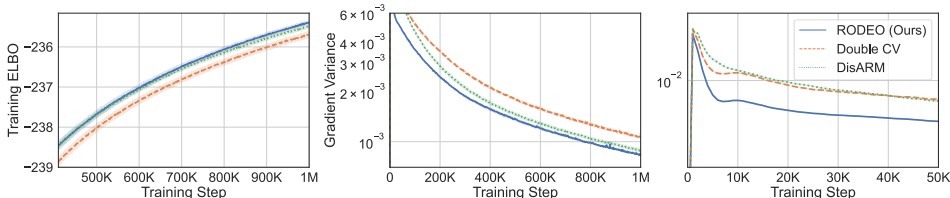

Figure 3: Training hierarchical binary latent VAEs with four stochastic layers on Fashion-MNIST. In this experiment, the estimators have very different behaviors towards the beginning and the end of training. We show this on the right by zooming into the first 50K steps of the gradient variance plot.

We plot the results on VAEs with four stochastic layers on Fashion-MNIST in Figure 3. The results for other datasets and for 2 and 3 stochastic layers can be found in Appendix E. RODEO generally achieves the best training ELBOs. One difference we noticed when comparing the variance results with those obtained from single-stochastic-layer VAEs is that these estimators have very different behaviors towards the beginning and the end of training. For example, in Figure 3, Double CV starts with lower variance than DisARM, but the gap diminishes after around 100K steps, and DisARM start to perform better as the training proceeds. In contrast, RODEO has the lowest variance in both phases for all datasets save Omniglot, where DisARM overtakes it in the long run.

## 6.3 Ablation study

Finally, we conduct an ablation study to gain more insight into the impact of each component of RODEO. In each experiment we train binary latent VAEs with $K = 2$.

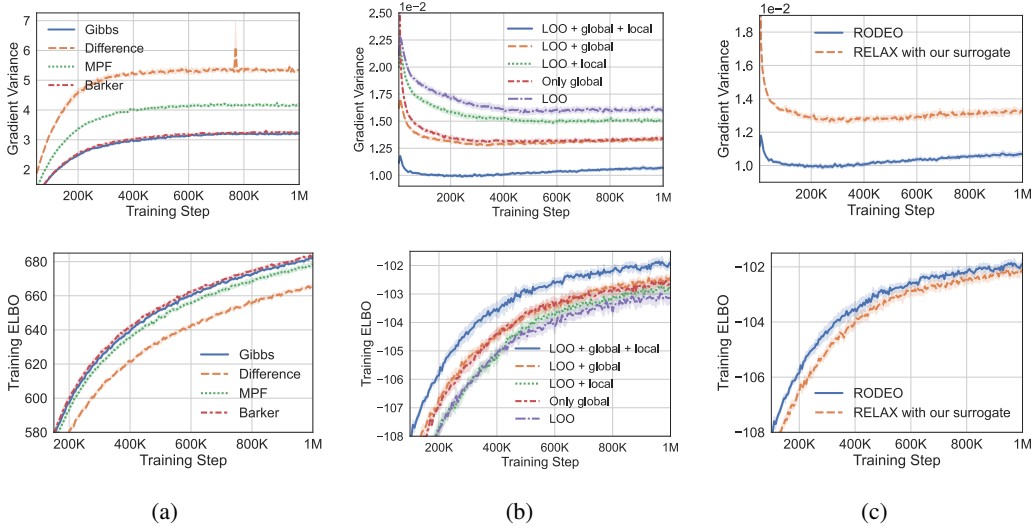

Figure 4: Ablation study of impact of RODEO components: (a) Stein operators, (b) LOO baseline and global and local CVs, (c) surrogate functions on binary VAE training performance.

**Impact of Stein operator**   Figure 4a explores the impact of RODEO Stein operator choice on non-binarized MNIST. As expected, the less stable difference (8) and MPF (6) operators lead to significantly higher gradient variances and worse training ELBOs. In fact, the same operators led to divergent training on binarized MNIST. The Barker operator (6) with neighborhoods defined by having one differing coordinate yields results very similar to Gibbs (4) as the operators themselves are very similar for binary variables (notice that $\frac{q(y)}{q(y)+q(x)} = q(y_i|x_{-i})$ when $x_{-i} = y_{-i}$).

**Impact of LOO baseline and global and local CVs**   Figure 4b explores binarized MNIST performance when RODEO is modified to retain a) only the LOO baseline (this is equivalent to standard RLOO), b) only the global CV, c) the LOO baseline + the global CV, or d) the LOO baseline + the local CV. We observe that the global and local Stein CVs both contribute to variance reduction and have complementary effects. Moreover, remarkably, the global Stein CV alone outperforms RLOO.

**Impact of surrogate functions**   To tease apart the benefits of our new surrogate functions (13) and the remaining RODEO components, we replace the surrogate function $c_\phi(z)$ in RELAX [20] with our surrogates, which only requires a single evaluation of $f$ per $x^{(k)}$ (see Appendix A for more details). Figure 4c compares the performance of RODEO and this modified RELAX on binarized MNIST. Since the surrogate functions are matched, the consistent improvements over modified RELAX can be attributed to the Stein and double CV components of RODEO. In Appendix C.2, we also experiment with increasing the complexity of $H$ and $H^*$ by using two hidden layers instead of a single hidden layer in the MLP. We did not observe significant improvements in variance reduction.

## 7   Conclusions, Limitations, and Future Work

This work tackles the gradient estimation problem for discrete distributions. We proposed a variance reduction technique which exploits Stein operators to generate control variates for REINFORCE leave-one-out estimators. Our RODEO estimator does not rely on continuous reparameterization of the distribution, requires no additional function evaluations per sample point, and can be adapted online to learn very flexible control variates parameterized by neural networks.

One potential drawback of our surrogate function constructions (13) and (14) is the need to evaluate an auxiliary function ($H$ or $H^*$) at $K-1$ locations. This cost can be comfortably borne when $H$ and $H^*$ are much cheaper to evaluate than $f$, such as in the ResNet VAE example in Appendix C.1 and many large VAE models used in practice [63]. And, in our experiments with the most common sample sizes, the runtime of RODEO was no worse than that of RELAX. To obtain a more favorable cost for large $K$, one could employ alternative surrogates that require only a constant number of

auxiliary function evaluations, e.g.,

$$h_k(y) = H\big(\tfrac{1}{K-1}\sum_{j\neq k} f(x^{(j)}), \tfrac{1}{K-1}\sum_{j\neq k}\nabla f(x^{(j)})^\top(y - x^{(j)})\big).$$

The runtime of RODEO could also be improved by varying the Stein operator employed. For example, the Gibbs operator $(Ah)(x)$ (4) used in our experiments evaluated its surrogate function $h$ at $d$ neighboring locations of $x$. This evaluation complexity could be reduced by subsampling neighbors, resulting in a cheaper but still valid Stein operator, or by employing the numerically stable Barker operator (6) with fewer neighbors. Either strategy would introduce a speed-variance trade-off worthy of study in follow-up work. Finally, we have restricted our focus in this work to differentiable target functions $f$. In future work, this limitation could be overcome by designing effective surrogate functions that make no use of derivative information.

## Acknowledgments and Disclosure of Funding

We thank Heishiro Kanagawa for suggesting appropriate names for the Barker Stein operator.

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
