## A  Sample-dependent Baselines in REBAR and RELAX

We start with the REINFORCE estimator with the sample-dependent baseline $b_k$:

$$\frac{1}{K}\sum_{k=1}^{K}(f(x^{(k)}) - b_k)\nabla_\eta \log q_\eta(x^{(k)}) + \mathbb{E}[b_k\nabla_\eta \log q_\eta(x^{(k)})]. \tag{15}$$

REBAR [62] introduces indirect independence on $x^{(k)}$ in $b_k$ through the continuous reparameterization $x = H(z), z^{(k)} \sim q_\eta(z|x = x^{(k)})$, where $z$ is a continuous variable and $H$ is an argmax-like thresholding function. Specifically, $b_k = f(\sigma_\lambda(z^{(k)}))$, where $\sigma_\lambda$ is a continuous relaxation of $H$ controlled by the parameter $\lambda$. The correction term decomposes into two parts:

$$\mathbb{E}_{x^{(k)}}[\mathbb{E}_{z^{(k)}|x^{(k)}}[f(\sigma_\lambda(z^{(k)}))]\nabla_\eta \log q_\eta(x^{(k)})]$$
$$= \nabla_\eta\mathbb{E}_{q_\eta(z)}[f(\sigma_\lambda(z))] - \mathbb{E}_{x^{(k)}}[\nabla_\eta\mathbb{E}_{q_\eta(z^{(k)}|x^{(k)})}[f(\sigma_\lambda(z^{(k)}))]].$$

Both parts can be estimated with the reparameterization trick [29, 47, 58] which often has low variance. The RELAX [20] estimator generalizes REBAR by noticing that $f(\sigma_\lambda(z))$ can be replaced with a free-form differentiable function $c_\phi(z)$. However, RELAX still relies on parameterizing $c_\phi(z)$ as $f(\sigma_\lambda(z)) + r_\theta(z)$ to achieve strong performance, as noted in Dong et al. [14].

To form modified RELAX in Section 6.3, we replace $b_k = c_\phi(z^{(k)})$ with $b_k = h_k(\sigma_\lambda(z^{(k)}))$ for $h_k$ defined in (13).

## B  Proof of Unbiasedness of RODEO

Recall our estimator defined in RODEO is

$$\frac{1}{K}\sum_{k=1}^{K}[(f(x^{(k)}) - \frac{1}{K-1}\sum_{j\neq k}(f(x^{(j)}) + (Ah_j)(x^{(j)}))) \cdot \nabla_\eta \log q_\eta(x^{(k)}) + (A\tilde{h}_k^\star)(x^{(k)})]. \tag{16}$$

To show the unbiasedness, we compute its expectation under $q_\eta$ as

$$\frac{1}{K}\sum_{k=1}^{K}\mathbb{E}_{q_\eta}[f(x^{(k)})\nabla_\eta \log q_\eta(x^{(k)})]$$

$$- \frac{1}{K(K-1)}\sum_{k=1}^{K}\sum_{j\neq k}\mathbb{E}_{q_\eta}[(f(x^{(j)}) + (Ah_j)(x^{(j)}))\nabla_\eta \log q_\eta(x^{(k)})]$$

$$+ \frac{1}{K}\sum_{k=1}^{K}\mathbb{E}_{q_\eta}[(A\tilde{h}_k^\star)(x^{(k)})].$$

Since the first term is the desired gradient $\nabla_\eta\mathbb{E}_{q_\eta}[f(x)]$ and the third term is zero, it suffices to show that the second term also vanishes. Using the law of total expectations, we find for $j \neq k$,

$$\mathbb{E}_{q_\eta}[(f(x^{(j)}) + (Ah_j)(x^{(j)}))\nabla_\eta \log q_\eta(x^{(k)})]$$
$$= \mathbb{E}_{x^{(k)}\sim q_\eta}[\mathbb{E}_{q_\eta}[f(x^{(j)}) + (Ah_j)(x^{(j)}) \mid x^{(k)}]\nabla_\eta \log q_\eta(x^{(k)})]$$
$$= \mathbb{E}_{x^{(k)}\sim q_\eta}[\mathbb{E}_{q_\eta}[f(x^{(j)}) \mid x^{(k)}]\nabla_\eta \log q_\eta(x^{(k)})]$$
$$= \mathbb{E}_{q_\eta}[f(x^{(j)})\nabla_\eta \log q_\eta(x^{(k)})]$$
$$= \mathbb{E}_{x^{(j)}\sim q_\eta}[f(x^{(j)})\mathbb{E}_{x^{(k)}\sim q_\eta}[\nabla_\eta \log q_\eta(x^{(k)}) \mid x^{(j)}]] = 0,$$

which completes the proof.

## C  Additional Experiments

### C.1  Wall clock time comparison with RLOO

Besides necessary target function evaluations, the RODEO estimator comes with the additional cost of evaluating the neural network-based $H, H^*$. Therefore, RODEO is most suited to the problems

Table 3: Architecture of the ResNet VAE in Appendix C.1. 3x3x$C$ means kernel size 3x3 and $C$ output channels. Each (De)conv Res block is composed of two (de)convolutional layers with strides 1, same padding, and ReLU activations, plus a skip connection with identity map. For Res blocks with downsample and upsample functions, the first convolutional layer has strides 2, and the skip connection is replaced by a convolutional layer with 2x2 kernel size and strides 2.

| Encoder | Decoder |
|---|---|
| Conv 3x3x16, strides 1, padding 1 | Fully connected, 7x7x64 units |
| Conv Res block 3x3x16 | Deconv Res block 3x3x64 |
| Conv Res block 3x3x16 | Deconv Res block 3x3x64 |
| Conv Res block 3x3x32 (downsample by 2) | Deconv Res block 3x3x32 (upsample by 2) |
| Conv Res block 3x3x32 | Deconv Res block 3x3x32 |
| Conv Res block 3x3x64 (downsample by 2) | Deconv Res block 3x3x16 (upsample by 2) |
| Conv Res block 3x3x64 | Deconv Res block 3x3x16 |
| Fully connected, 200 units | Deconv 3x3x1, strides 1, padding 1 |

where the cost of evaluating $f$ dominates that of evaluating $H, H^*$. This is often the case in practice. For example, state-of-the-art variational autoencoders [e.g., 63] are often built on expensive neural architectures such as deep residual networks (ResNets). Here, to demonstrate the practical advantage of our method as the complexity of $f$ grows, we replace the two-layer MLP VAEs used in previous experiments with a ResNet VAE (architecture shown in Table 3), while the neural network used by $H, H^*$ remains a single-layer MLP with 100 hidden units. We then compare the wall clock performance of RODEO with RLOO. The latent variables in this experiment remain binary and have 200 dimensions.

The results are shown in Figure 5. RODEO achieves better training ELBOs than RLOO in the same amount of time. In fact, for this VAE architecture, the per-iteration time of RODEO is **25.2ms**, which is very close to the **23.1ms** of RLOO. This indicates that the cost of $f$ is significantly higher than that of $H, H^*$.

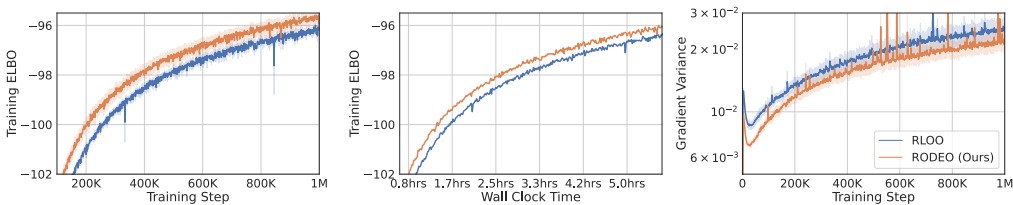

Figure 5: Comparing the performance of RODEO and RLOO on more expensive ResNet VAE models trained on binarized MNIST with $K = 2$. The middle plot shows the average wall clock performance over 5 trials.

## C.2 Impact of neural network architectures of surrogate functions

We conduct one more ablation study to investigate the impact of neural network architectures used by $H, H^*$. Specifically, we replace the single-hidden-layer control variate network used in previous experiments with a two-hidden-layer MLP (each layer has 100 units) and compare their performance on binarized MNIST with $K = 2$. We keep other settings the same as in Section 6.1. The results are plotted in Figure 6. We do not observe significant difference between the two versions of RODEO.

# D  Experimental Details

Our implementation is based on the open-source code of DisARM [14] (Apache license) and Double CV [60] (MIT license). Our figures display 1M training steps and our tables report performance after 1M training steps to replicate the experimental settings of DisARM [14].

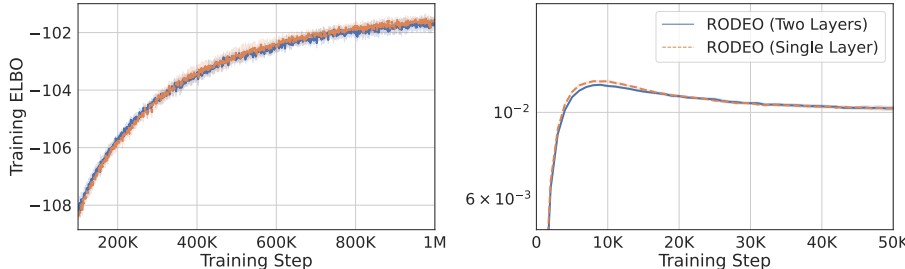

Figure 6: Comparing the performance of RODEO with single-hidden-layer and two-hidden-layer neural network architectures for $H, H^*$ on binary VAEs.

## D.1 Details of VAE experiments

VAEs are models with a joint density $p(y, x) = p(y|x)p(x)$, where $x$ denotes the latent variable. $x$ is assigned a uniform factorized Bernoulli prior. The likelihood $p_\theta(y|x)$ is parameterized by the output of a neural network with $x$ as input and parameters $\theta$. The VAE has two hidden layers with 200 units activated by LeakyReLU with the coefficient 0.3. To optimize the VAE we use Adam with base learning rate $10^{-4}$ for non-binarized data and $10^{-3}$ for dynamically binarized data, except for binarized Fashion-MNIST we decreased the learning rate to $3 \times 10^{-4}$ because otherwise the training is very unstable for all estimators. We use Adam with the same learning rate $10^{-3}$ for adapting our control variate network in all experiments. The batch size is 100. The settings of other estimators are kept the same with Titsias and Shi [60].

In the minimum probability flow (MPF) and Barker Stein operator (6), we choose the neighborhood $\mathcal{N}_x$ to be the states that differ in only one coordinate from $x$. Let $y \in \mathcal{N}_x$ be an element in this neighborhood such that $y_i \neq x_i$ and $y_{-i} = x_{-i}$. For the MPF Stein estimator and the difference Stein estimator (8), the density ratio $\frac{q(y)}{q(x)}$ can be simplified to $\frac{q(y_i|x_{-i})}{q(x_i|x_{-i})}$. We further replace it with $\frac{q(y_i|x_{-i})}{q(x_i|x_{-i})+10^{-3}}$ to alleviate numerical instability. The Barker Stein estimator does not suffer from the numerical issue since the coefficient is bounded in the Bernoulli case: $\frac{q(y)}{q(x)+q(y)} = \frac{q(y_i|x_{-i})}{q(x_i|x_{-i})+q(y_i|x_{-i})} = q(y_i|x_{-i})$. In our experiments, we find that the difference Stein estimator is highly unstable and may diverge as the iteration proceeds.

## D.2 Details of hierarchical VAE experiments

We optimize the hierarchical VAE using Adam with base learning rate $10^{-4}$. Our control variate network is optimized using Adam with learning rate $10^{-3}$. Settings of training multilayer VAEs are kept the same with Dong et al. [14], except that we do not optimize the prior distribution of the VAE hidden layer and use a larger batch size 100.

# E Additional Results

In this section, we measure test set performance using 100 test points and the marginal log-likelihood bound of Burda et al. [10], which provides a tighter estimate of marginal log likelihood than the ELBO. Throughout, we call this the "test log-likelihood bound."

Table 4: Average 100-point test log-likelihood bounds of binary latent VAEs trained with $K = 2, 3$ (except for RELAX which uses 3 evaluations per step) on MNIST, Fashion-MNIST, and Omniglot. We report the average value $\pm 1$ standard error after 1M steps over 5 independent runs.

| | | Bernoulli Likelihoods | | | Gaussian Likelihoods | | |
|---|---|---|---|---|---|---|---|
| | | MNIST | Fashion-MNIST | Omniglot | MNIST | Fashion-MNIST | Omniglot |
| $K = 2$ | DisARM | $-101.61 \pm 0.07$ | $-239.11 \pm 0.11$ | $-118.34 \pm 0.05$ | $669.26 \pm 0.53$ | $163.40 \pm 0.59$ | $305.32 \pm 0.89$ |
| | Double CV | $-100.91 \pm 0.04$ | $-239.00 \pm 0.17$ | $-118.45 \pm 0.09$ | $677.02 \pm 0.93$ | $164.99 \pm 0.71$ | $304.72 \pm 1.39$ |
| | RODEO (Ours) | $\mathbf{-100.78 \pm 0.16}$ | $\mathbf{-238.97 \pm 0.09}$ | $\mathbf{-118.09 \pm 0.05}$ | $\mathbf{681.11 \pm 0.31}$ | $\mathbf{168.26 \pm 0.73}$ | $\mathbf{308.55 \pm 1.02}$ |
| $K = 3$ | ARMS | $-99.08 \pm 0.12$ | $-238.19 \pm 0.11$ | $-116.78 \pm 0.13$ | $688.61 \pm 0.84$ | $174.14 \pm 0.44$ | $320.45 \pm 1.07$ |
| | Double CV | $-99.16 \pm 0.12$ | $-238.54 \pm 0.16$ | $-116.75 \pm 0.15$ | $690.28 \pm 0.49$ | $173.67 \pm 0.30$ | $322.88 \pm 1.10$ |
| | RODEO (Ours) | $\mathbf{-98.72 \pm 0.14}$ | $\mathbf{-237.97 \pm 0.12}$ | $\mathbf{-116.69 \pm 0.09}$ | $\mathbf{695.11 \pm 0.33}$ | $\mathbf{174.57 \pm 0.30}$ | $\mathbf{323.92 \pm 1.24}$ |
| | RELAX (3 evals) | $-100.80 \pm 0.09$ | $-239.03 \pm 0.11$ | $-117.60 \pm 0.06$ | $686.21 \pm 0.57$ | $171.43 \pm 0.61$ | $317.78 \pm 1.25$ |

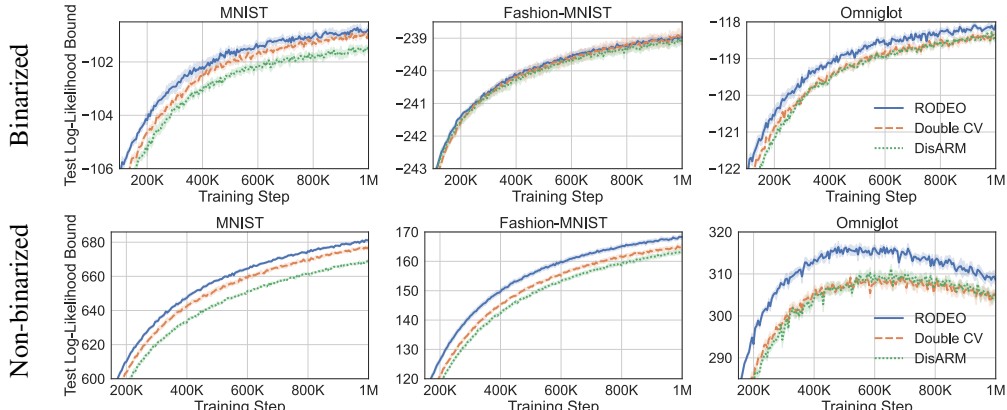

Figure 7: Average 100-point test log-likelihood bounds for binary latent VAEs trained on (top) dynamically binarized and (bottom) non-binarized MNIST, Fashion-MNIST, and Omniglot using $K = 2$.

Table 5: Average running time across $10^4$ steps on an NVIDIA 3080Ti GPU with an AMD 5950X CPU for the VAE experiment on binary MNIST in Section 6.1.

| | Double CV | DisARM/ARMS | RODEO (Ours) | RELAX (3 evals) |
|---|---|---|---|---|
| $K = 2$ | 2.11 ms/step | 1.89 ms/step | 3.08 ms/step | 4.71 ms/step |
| $K = 3$ | 2.28 ms/step | 1.91 ms/step | 4.72 ms/step | |

Table 6: Average running time across $10^4$ steps on an NVIDIA 3080Ti GPU with an AMD 5950X CPU when training hierarchical VAEs with $K = 2$.

| | Double CV | DisARM | RODEO (Ours) |
|---|---|---|---|
| Two layers | 4.33 ms/step | 3.54 ms/step | 6.79 ms/step |
| Three layers | 7.69 ms/step | 6.09 ms/step | 10.61 ms/step |
| Four layers | 11.67 ms/step | 9.53 ms/step | 14.91 ms/step |

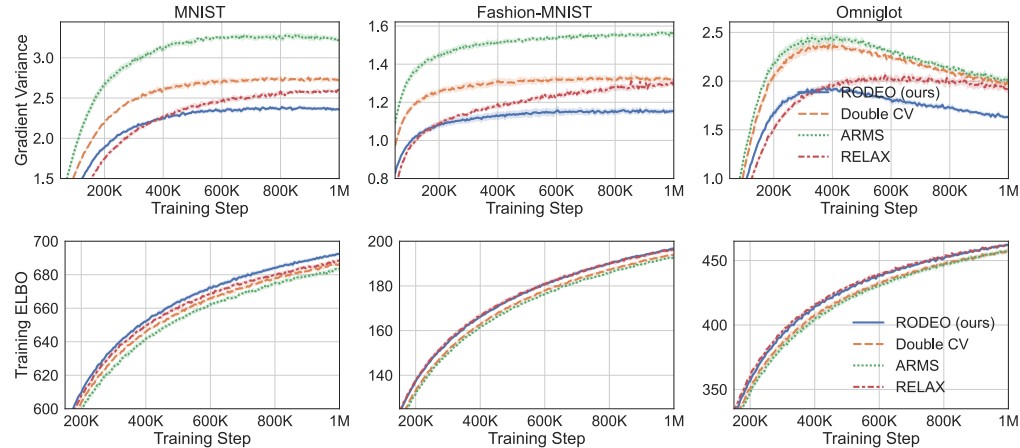

Figure 8: Training binary latent VAEs with Gaussian likelihoods with three evaluations of $f$ per step using RODEO/Double CV/ARMS with $K = 3$ or RELAX on non-binarized MNIST, Fashion-MNIST, and Omniglot. (Top) variance of gradient estimates. (Bottom) the plot of average ELBO on training examples against training steps.

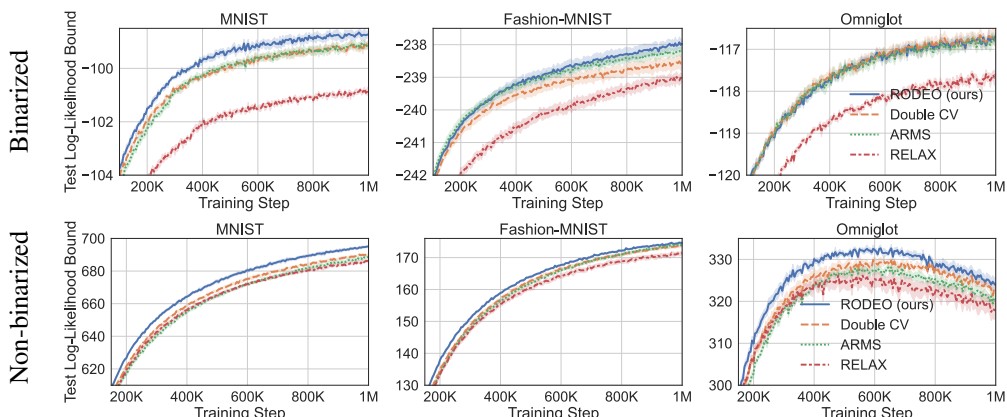

Figure 9: Average 100-point test log-likelihood bounds for binary latent VAEs trained on (top) dynamically binarized and (bottom) non-binarized MNIST, Fashion-MNIST, and Omniglot with three evaluations of $f$ per step using RODEO/Double CV/ARMS with $K = 3$ or RELAX.

Table 7: Training hierarchical binary latent VAEs on dynamically binarized MNIST, Fashion-MNIST, and Omniglot. We report the average ($\pm 1$ standard error) training ELBOs and 100-point test log-likelihood bounds after 1M steps over 5 independent runs.

|  |  | Training ELBO | | | Test Log-Likelihood Bound | | |
|---|---|---|---|---|---|---|---|
|  |  | MNIST | Fashion-MNIST | Omniglot | MNIST | Fashion-MNIST | Omniglot |
| Two layers | Double CV | $-103.52 \pm 0.06$ | $-239.82 \pm 0.07$ | $-114.06 \pm 0.04$ | $-97.62 \pm 0.08$ | $-237.65 \pm 0.06$ | $-110.48 \pm 0.04$ |
|  | DisARM | $-103.39 \pm 0.12$ | $\mathbf{-239.67 \pm 0.06}$ | $\mathbf{-113.67 \pm 0.05}$ | $-97.56 \pm 0.07$ | $\mathbf{-237.61 \pm 0.06}$ | $-110.15 \pm 0.04$ |
|  | RODEO (Ours) | $\mathbf{-103.15 \pm 0.07}$ | $-239.76 \pm 0.09$ | $-113.84 \pm 0.11$ | $\mathbf{-97.43 \pm 0.03}$ | $-237.63 \pm 0.07$ | $-110.32 \pm 0.10$ |
| Three layers | Double CV | $-97.59 \pm 0.15$ | $-234.34 \pm 0.07$ | $-108.66 \pm 0.06$ | $-93.71 \pm 0.12$ | $-234.34 \pm 0.07$ | $-107.48 \pm 0.07$ |
|  | DisARM | $-97.95 \pm 0.30$ | $-234.45 \pm 0.05$ | $-108.60 \pm 0.08$ | $-94.12 \pm 0.28$ | $-234.46 \pm 0.06$ | $-107.32 \pm 0.10$ |
|  | RODEO (Ours) | $\mathbf{-97.21 \pm 0.17}$ | $\mathbf{-234.11 \pm 0.10}$ | $\mathbf{-108.51 \pm 0.04}$ | $\mathbf{-93.52 \pm 0.16}$ | $\mathbf{-234.19 \pm 0.07}$ | $\mathbf{-107.26 \pm 0.06}$ |
| Four layers | Double CV | $-98.73 \pm 0.06$ | $-235.69 \pm 0.07$ | $-110.92 \pm 0.06$ | $-93.28 \pm 0.03$ | $-234.63 \pm 0.03$ | $-107.86 \pm 0.03$ |
|  | DisARM | $-98.97 \pm 0.02$ | $-235.50 \pm 0.04$ | $-110.85 \pm 0.07$ | $-93.56 \pm 0.04$ | $-234.52 \pm 0.04$ | $-107.87 \pm 0.05$ |
|  | RODEO (Ours) | $\mathbf{-98.67 \pm 0.14}$ | $\mathbf{-235.39 \pm 0.05}$ | $\mathbf{-110.79 \pm 0.03}$ | $\mathbf{-93.27 \pm 0.09}$ | $\mathbf{-234.39 \pm 0.06}$ | $\mathbf{-107.77 \pm 0.02}$ |

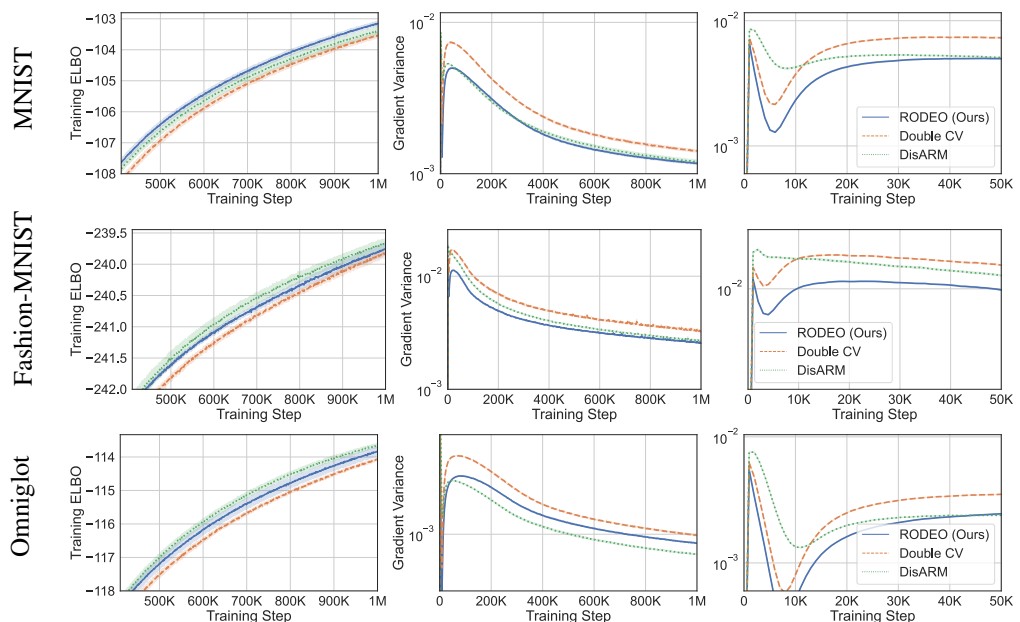

Figure 10: Training hierarchical binary latent VAEs with **two** stochastic layers on dynamically binarized MNIST, Fashion-MNIST and Omniglot. We plot (left) the average ELBO on training examples and (middle) variance of gradient estimates. We zoom into the first 50K steps of the variance plot on the right figure.

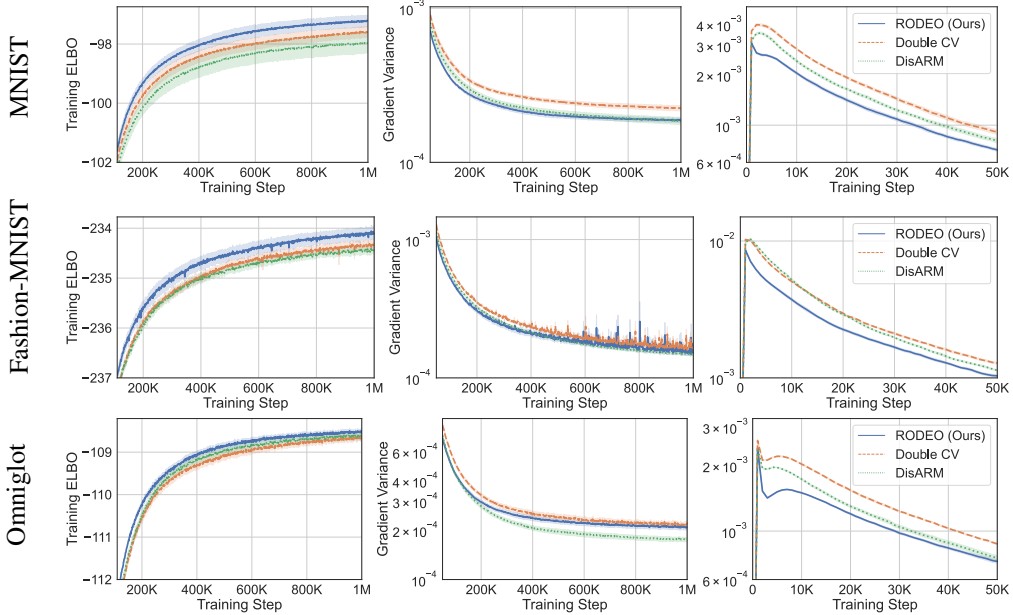

Figure 11: Training hierarchical binary latent VAEs with **three** stochastic layers. on dynamically binarized MNIST, Fashion-MNIST and Omniglot. We plot the average ELBO on training examples (left) and variance of gradient estimates (middle). We zoom into the first 50K steps of the variance plot on the right figure.

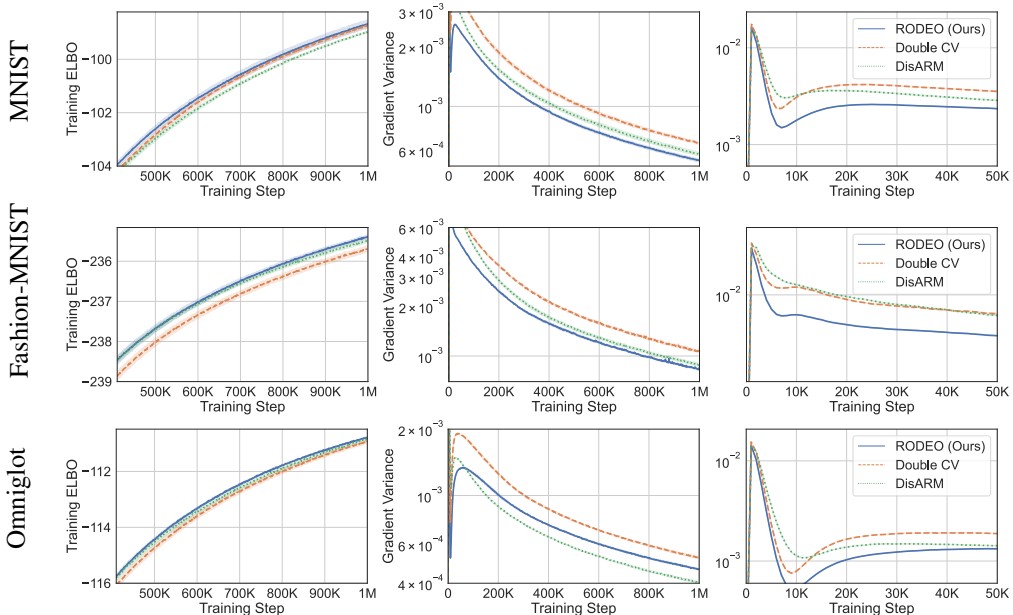

Figure 12: Training hierarchical binary latent VAEs with **four** stochastic layers on dynamically binarized MNIST, Fashion-MNIST and Omniglot. We plot the average ELBO on training examples (left) and variance of gradient estimates (middle). We zoom into the first 50K steps of the variance plot on the right figure.