# OpenReview forum: "Gradient Estimation with Discrete Stein Operators"
_NeurIPS.cc/2022/Conference — NeurIPS 2022 Accept_

### Official Review · Reviewer_K2dM · 2022-07-08

**Rating:** 8
**Confidence:** 4
**Soundness:** 4 excellent
**Presentation:** 4 excellent
**Contribution:** 4 excellent

**Summary:**

The paper introduces discrete Stein operators for gradient estimation applied to Bernoulli variational autoencoders. As is widely known, gradients play a crucial role in training machine learning algorithms and taking gradients with respect to a discrete variable is non-trivial. Existing approaches tend to produce gradients with a high variance which then requires control variate approaches to manage the variance of the gradient. In this paper, the authors address this problem with a variance reduction technique based on Stein operators for discrete distributions.

The authors test the efficacy of their approach on three variations of the binary latent variable VAE. In the experiments, the authors show that their discrete Stein operator is competitive with existing state-of-the-art approaches.

**Questions:**

- In the algorithm box $x^{(1:K)} \sim q_\eta$ are iid. Is this necessary or can they be generated from a Markov chain?
- Is requiring $\nabla_\theta f(x)$ a restriction of this approach compared to the other approaches discussed in the paper?
- It feels like the choice of the function $H$ (i.e. eq. 16) is important for the efficiency of the control variate. Can more be said about the choice of the neural architecture in H?
- What is the computational cost for the algorithms in Table 2?
- The gradient $g_gamma$ isn't clearly defined.


**Limitations:**

The limitations of the approach are discussed in the conclusion section. There are no obvious negative societal issues.

**Strengths And Weaknesses:**

Originality - The paper proposes a new approach to variance-reduced gradient estimation for discrete variables using Stein operators. As far as I am aware, this is a novel contribution to the research area. A potential weakness in terms of originality is perhaps that the main benefit of this approach, in terms of its efficiency, is likely to stem from the variance-reduction aspect, which is based on the reinforce leave-one-out (RLOO) estimator and a critical perspective might suggest that this paper is taking RLOO and plugging it into a Stein operator, which may reduce the novelty slightly.
Quality - The paper is of high quality and well-placed in the NeurIPS community.
Clarity - The paper is very well written and the methodology is clearly presented. There are no obvious weaknesses in the presentation that the authors could aim to improve.
Significance - The paper nicely addresses the existing literature in the field which helps to place this work in the wider field. The problem addressed in the paper is significant in the ML community and the contribution of this paper is significant for both its contribution to latent variable VAEs and Stein method. In theory, the paper could be applied to general discrete variables, but in the paper, only VAE-type examples are considered. Is this a limitation of the approach?

---

> ### Author Response · Authors · 2022-08-02
> **Response**
>
> Thank you for the time you’ve taken to review our work and for the positive and constructive feedback! We are glad that you found our work of high quality, well-placed in the NeurIPS community, well written and presented, and significant.  We respond to each of your remaining comments below.
>
>
> ### Novelty
> We agree that the variance reduction effect of our estimator has contributions from RLOO. On the other hand, our ablation study showed that all components of RODEO (RLOO, global and local Stein control variates) have complementary effects in variance reduction.
>
> ### Applicable to general discrete variables
> We agree with the reviewer that our approach is applicable to general discrete variables and is not limited to VAEs. It can be applied to any problem that requires optimizing the expectation with respect to the underlying distribution parameters, and we experiment with VAEs because they are the standard testbed for such algorithms in prior literature. In addition, our discrete Stein operators can be applied to even broader contexts, such as variance reduction for Monte Carlo estimation.
>
>
> ### Can x(1:K) be generated from a Markov chain?
> In principle you can substitute sample points from a Markov chain targeting $q_\eta$ at the cost of a bias penalty in the gradient estimation.  Any other sampling scheme that yields an unbiased gradient estimate (e.g., importance sampling) can be substituted as well.  We should also highlight that in the applications driving this work, sampling directly from $q_\eta$ is fast and inexpensive.
>
> ### Is requiring $\nabla_\theta f(x)$ a restriction?
> We clarify that we do not require $\nabla_\theta f(x)$ but $\nabla f(x)$ for our surrogate function design. For other surrogate functions this requirement can be removed. The reason why we include $\nabla_\theta f(x)$ in the algorithm is that in many applications, including VAEs, the target function $f$ also has parameters $\theta$ that need to be optimized. In this case, we can reuse the same backpropagation to get $\nabla f(x)$. For other applications, this can be a restriction compared to RLOO and DisARM which do not require gradients of $f$.
>
> ### Choice of the neural architecture in H?
> As stated in line 217, “The functions H (16) and H^* (18) share a neural network architecture with two 2 output units and a single hidden layer with 100 units”. We provide evidence in Appendix C.1 that, for commonly used VAE architectures (different from the ones used as benchmarks in main text), the cost of $f$ dominates that of $H,H^*$. In such cases, the per-iteration cost of RODEO is very close to RLOO.
>
> ### What is the computational cost for the algorithms in Table 2?
> Let the cost of evaluating $f$ be C(f) (we include the cost of both forward pass and backpropagation since they are both required by applications like VAEs). Then the cost for RLOO, DisARM, Double CV is KC(f), the cost for RELAX with the default surrogate is 3KC(f) and the cost for RODEO is K(C(f) + (K-1)*C(g)), where g is the network shared by $H, H\^\*$. The RODEO estimator is favored when $f$ is expensive compared to $g$ and $K$ is small (e.g., K=2), which is often the case in practice (see Appendix C.1 in the revision where the runtime of RODEO becomes comparable to RLOO for more complex architectures like ResNets).
>
> ### The gradient $g_\gamma$ isn't clearly defined.
> We apologize for the omission and have revised the main text to explicitly define $g_\gamma$ as equation (17).

---

### Official Review · Reviewer_34qn · 2022-07-10

**Rating:** 7
**Confidence:** 4
**Soundness:** 3 good
**Presentation:** 4 excellent
**Contribution:** 3 good

**Summary:**

This paper studies the design of control variates for REINFORCE-type gradient estimators in discrete expectation optimization problems. The paper proposes a general class of control variates using discrete Stein operators, and a practical unbiased gradient estimator RODEO with Stein-inspired control variates. Experiments on Bernoulli VAEs show the proposed method achieves lower gradient variance and better performance compared to a variety of baselines.

**Questions:**

There are no major questions the reviewer would like to raise. There are 2 minor questions of interest:

- Equation (14) is built under the intuition of an “optimal” Markov chain, but in practice the Markov chain is clearly not optimal with limited transition neighborhood especially in high dimensions. In the latter case, it would be helpful to analyze its implications for (14). Alternatively, it would be helpful to study the amount of variance reduction given by Rao-Blackwellization when $h=f$ exactly as dimension increases?
- On page 7, it is mentioned that “RELAX was often observed to have very strong performance in prior work”. Why is it comparatively worse than all baselines in this paper?


**Limitations:**

The authors adequately addressed the limitations of their work and proposed interesting future directions of study. The authors did not discuss any potential negative societal impacts of the work, but as the work is mostly theoretical it is unlikely to have immediate negative societal impacts.

**Strengths And Weaknesses:**

Strengths:

- The paper is well-written and gives a clear introduction to previous variance reduction methodologies of relevance.
- The theory is well-substantiated and well-explained. The simplifications made in (14) and (16) are justified, and the intuition that $h$ serves as a surrogate for $f$ is clearly built by relating to Rao-Blackwellization.
- A number of Stein operators are cited, with practical guidelines for choosing suitable operators based on numerical stability.
- Experiment results are comprehensive and inclusive of the best introduced baseline methods. The comparisons are performed fairly under similar computational costs, and the slightly increased cost for RODEO due to evaluating auxiliary functions is clearly highlighted and justified. Ablation studies clearly demonstrate the effect of different components of the proposed method.

Weaknesses:

- While the proposed methodology is original and novel, similar ideas based on Stein’s methods exist for REINFORCE-type gradients of continuous distributions and as general control variates for evaluating expectations. Nonetheless, this has been clearly detailed in the related works section.
- The method is evaluated only using VAE with binary latents, which has similar problem structure across different experiments, and may or may not favor the proposed methodology. In particular, the binary state space is relatively simplistic, albeit in a relatively high dimension of 200.

---

> ### Author Response · Authors · 2022-08-02
> **Response**
>
> Thank you for taking the time to review our work and for the positive and constructive feedback!  We are glad that you found our paper well-written, our theory well-substantiated, and our results comprehensive. We respond to each of your remaining comments below.
>
> ### Stein’s methods exist for REINFORCE-type gradients of continuous distributions and as general control variates for evaluating expectations
> Applying Stein’s methods to discrete gradient estimation problems is highly nontrivial due to: 1) the lack of numerically stable discrete Stein operators; 2) the higher complexity compared to continuous Stein operators. We made significant contributions to both by introducing numerically stable Gibbs and Barker Stein operators and a novel design of the surrogate function to avoid additional evaluation of target function.
>
> ### Evaluations have similar problem structure, i.e., VAE with binary latents:
> We designed our experiments to reflect many possible changes of the problem structure: 1) smaller and larger scale of $f$, using Bernoulli and Gaussian likelihoods, 2) different datasets, and 3) dependence of stochasticity, using multiple layers of latent variables. We agree with the reviewer that studying the method on non-binary discrete distributions could provide additional insights. Nevertheless, we believe that deserves a whole piece of future work.
>
> ### Optimal i.i.d. Markov chain and Implications for (14). The amount of variance reduction given by Rao-Blackwellization when h=f as dimension increases:
> We consider the optimal i.i.d. chain to draw inspiration about the form of $\tilde{h}$, for which we do not expect theoretical implications. The theoretical justification is given by the Rao-Blackwellization when $h=f$. The amount of variance reduction is $E[Var[f(X_{t+1})\nabla_{\eta_i} \log q_\eta(X_{t+1})|X_t]]$ according to the law of total variance. It heavily depends on the form of $f$ and the choice of the chain. Studying how that relates to the number of dimensions requires making assumptions about how $f$ changes with dimensions of inputs.
>
> ### why RELAX is comparatively worse than all baselines in this paper:
> Prior work (e.g., DisARM) usually compares RELAX with the $K=2$ version of baselines. However, this is not a fair comparison because RELAX with the default surrogate function requires three evaluations of $f$ per iteration while the baselines only uses two. We use $K=3$ for other methods in order to match the per-iteration cost of RELAX. The result clearly shows that we should prefer other methods over RELAX if we aim to evaluate $f$ three times per iteration.

---

> > ### Comment · Reviewer_34qn · 2022-08-07
> > **Thank you for your comments**
> >
> > Thank you very much for your helpful answers - they addressed my questions very clearly. I increased my confidence to 4 and recommend the acceptance of this paper.

---

### Official Review · Reviewer_5uM4 · 2022-07-11

**Rating:** 7
**Confidence:** 3
**Soundness:** 3 good
**Presentation:** 3 good
**Contribution:** 3 good

**Summary:**

In this work, the authors propose a new type of control variate (CV) for discrete distributions. The proposed method is based on Stein operators.
The authors further incorporate the double CV into this proposed framework.
Experiments are shown to support the claim.

**Questions:**

See the weaknesses section.

**Strengths And Weaknesses:**

Strengths:
* 1 A new type of control variate (CV) for discrete distributions.
* 2 The proposed method is sound.
* 3 The proposed method can be used to train deep NNs.
* 4 Many empirical results are given to support the claim.

Weaknesses:
* 1 Discuss whether the proposed method can be applied to a discrete distribution with infinite support such as a Poisson distribution.
* 2 The authors should discuss the iteration cost (computational budge) of the proposed method. It will be great if the authors discuss the iteration cost of all related methods including baseline methods.
* 3 The running time instead of the number of training steps should be included in at least one of the plots.

---

> ### Author Response · Authors · 2022-08-02
> **Response**
>
> Thank you for the time you’ve taken to review our work and for the overall positive and constructive feedback! We address each individual comment below.
>
> ### Discrete distribution with infinite support, e.g., Poisson:
> Yes, our method can be applied to discrete distributions with infinite support. As we note on line 120 for the difference Stein operator: "An analogous operator is available for countably infinite $\mathcal{X}$ [24]," which includes the Poisson distribution. Similarly, the numerically stable Barker Stein operator can be applied directly to distributions with infinite support, for example, by choosing the neighborhood of x as {inc(x), dec{x}}.
>
> ### Iteration cost:
> Let the cost of evaluating $f$ be C(f) (we include the cost of both forward pass and backpropagation since they are both required by applications like VAEs). Then the cost for RLOO, DisARM, Double CV is KC(f), the cost for RELAX with the default surrogate is 3KC(f) and the cost for RODEO is K(C(f) + (K-1)*C(g)), where g is the network shared by $H, H\^\*$. The RODEO estimator is favored when $f$ is expensive compared to $g$ and $K$ is small (e.g., K=2), which is often the case in practice (see Appendix C.1).
>
> ### Plot against running time instead of training steps:
> Thanks for the suggestion. In the latest draft we have included a wall clock time comparison of our method with RLOO in Appendix C.1. In this experiment we replace the two-layer MLP-based VAE with a standard ResNet architecture, where the cost of $f$ is significantly higher than the single-layer MLP of $H, H^*$. RODEO achieves better training ELBOs than RLOO in the same amount of time.

---

### Official Review · Reviewer_avva · 2022-07-13

**Rating:** 7
**Confidence:** 3
**Soundness:** 3 good
**Presentation:** 3 good
**Contribution:** 3 good

**Summary:**

This paper introduces a control variate scheme for REINFORCE-type estimators based on the Stein operators for discrete distributions. Discrete Stein operators generate zero-mean functions under the discrete sampling distribution; therefore, they can be used to construct control variates if a suitable function is chosen. Based on this insight, the authors propose RODEO, a CV scheme based on discrete Stein operators for the REINFORCE leave-one-out (RLOO) estimator. RODEO contains to adaptive function $H$ and $H^*$ that are optimised by minimising the variance of the estimator. RODEO is validated on binary VAE models and is shown to be effective in reducing the variance and inducing faster convergence.

**Questions:**

* What is the wall-clock time performance of this method compared to RLOO? I am especially interested in comparing its wall-clock performance compared to simply increasing the number of samples in RLOO. I know that this is dependent on the architecture of $H$ and $H^*$, but it is good to have some indication to its practical feasibility.
* Can this type of estimator benefit from a parallel architecture? For instance, running the model on one GPU and $H$ and $H^*$ on another? $H$ and $H^*$ seem to take $f$ and its gradients as inputs, so this might be challenging at a first glance. Can the authors comment on this point?
* When is it practically feasible to use this estimator? Are the extra implementation and computation overheads worth it compared to simply increasing the number of MC samples in RLOO or even vanilla REINFORCE? I think it good to add a discussion of its practicality to final section.
* What is the effect of the architecture of $H$ and $H^*$ on the variance reduction? Is bigger always better? And is there any trade-off in terms of computation vs variance reduction?

**Limitations:**

Some limitations of this work are addressed in the conclusion and their discussion was, in my opinion, sufficient. There is no potential negative societal impact in my assessment.

**Strengths And Weaknesses:**

To the best of my knowledge, the use of Stein operators in gradient estimation in discrete settings is novel. The authors attempt to place their work within the literature, and they make a sufficient argument for the originality of their contribution.

The development and explanation of the methodology is very clear, even for someone who is not previously familiar with Stein operators. That being said, I found the build-up leadin to equation (14) somewhat confusing. How does a Markov chain generate i.i.d samples? This section is important for developing the estimator, so I think the authors should improve the clarity of this derivation (or expand it in the appendix).

Overall, I think this is a good paper. There are no major concerns with regards to the motivation and the development of the idea. The empirical evaluation can be improved by further comparisons. e.g. vanilla RLOO with different number of samples, and an illustration of the limitation of this method when the number of samples increases (this point was acknowledged in the conclusion). It is lacking a discussion of the practicality of this estimator given its complexity.

In my assessment, the main strengths of this work are its originality and clarity in the motivation and the development of the methodology, as well as its potential impact on future research. Its main weaknesses are the limited evaluation and discussion on its practicality.

---

> ### Author Response · Authors · 2022-08-02
> **Response**
>
> Thank you for the time you’ve taken to review our work and for the positive and constructive feedback! We are glad that you found our paper original and potentially impactful on future research. We respond to each of your comments below.
>
> ### Motivation of equation (14)
> We apologize for the confusion. We will clarify that direct i.i.d. sampling from $q_\eta$ is a special case of a Markov chain (it's a Markov chain in which the current state is ignored entirely and the new state is generated independently from $q_\eta$).  However, for such a chain the correction term requires exact expectations under $q_\eta$, which is intractable. One can view the alternative chains that we employ in practice as tractable approximations to this “optimal” direct sampling Markov chain. We use the optimal chain to draw inspiration about the form of $\tilde{h}$, for which we also provide theoretical justification based on Rao-Blackwellization.
>
> ### Further comparisons with RLOO, wall clock time performance & practical feasibility
> We note that, for RODEO, as the complexity of $f$ grows, the cost of target function evaluations dominates the control variate construction overhead because the cost of $H, H^*$ remains constant. Following your suggestion, in the latest draft we have included a wall clock time comparison of our method with RLOO in Appendix C.1. In this experiment we replace the two-layer MLP-based VAE with a standard ResNet architecture, where the cost of $f$ is significantly higher than the single-layer MLP of $H, H^*$. In this case, RODEO and RLOO have very close per-iteration time (0.025s vs. 0.023s). And RODEO achieves better training ELBOs than RLOO for the same amount of time.
>
> Our estimator is practical and should be preferred over RLOO when $f$ is expensive to evaluate, such as in the ResNet VAE example in Appendix C.1. In such cases, K=2 is  the most relevant setting in practice–For large VAE models used in practice (e.g., Vahdat et al., 2020), evaluating $f$ K times for a large $K$ can be unaffordable. In this paper, we only study higher K to provide a fair comparison against RELAX (which uses more function evaluations per sample point than other methods). Moreover, we have shown that, when the cost of $f$ dominates $H,H*$, the overhead of RODEO estimator compared to RLOO can be negligible. We will add more discussion about practicality to the final section.
>
> ### Parallel architecture
> As the reviewer correctly notes, the current design of $H, H^*$ does not allow computing them in parallel with $f$ since they depend on $\nabla f(x)$ as inputs. However, we should note that our method is not really limited to such a surrogate function design, and there exist other choices that allow such parallel computation. Moreover, as our estimator is aimed at problems where the cost of $f$ dominates those of $H,H^*$, we believe this is not a drawback of our approach.
>
> ### Effect of the architecture of $H,H^*$
> In Appendix C.2 of the latest revision, we experimented with increasing the complexity of $H,H^*$, e.g., using two-layer MLP instead of single-layer. We did not observe significant improvements in variance reduction.
>
> Refs:
> Vahdat, A., & Kautz, J. (2020). NVAE: A deep hierarchical variational autoencoder. Advances in Neural Information Processing Systems, 33, 19667-19679.

---

> > ### Comment · Reviewer_avva · 2022-08-07
> > **Thank you for your response**
> >
> > I appreciate your detailed response to my questions and the changes made to the manuscript based on those. I think the paper is overall a good paper and I would recommend its acceptance.

---

### Meta-Review · Area_Chair_7vZi · 2022-08-27

**Recommendation:** Accept
**Confidence:** Certain

**Metareview:**

Thanks to the authors for this submission.  The reviewers all agreed that this is a novel and effective method for reducing gradient estimates in a discrete setting.  The reviewer-author discussion was fruitful, and changes to the manuscript have improved the submission.


**Award:**

Yes

---

### Decision · Program_Chairs · 2022-09-14

Accept